# Cognitive Crescendo: How Music Shapes the Brain’s Structure and Function

**DOI:** 10.3390/brainsci13101390

**Published:** 2023-09-29

**Authors:** Corneliu Toader, Calin Petru Tataru, Ioan-Alexandru Florian, Razvan-Adrian Covache-Busuioc, Bogdan-Gabriel Bratu, Luca Andrei Glavan, Andrei Bordeianu, David-Ioan Dumitrascu, Alexandru Vlad Ciurea

**Affiliations:** 1Department of Neurosurgery, “Carol Davila” University of Medicine and Pharmacy, 020021 Bucharest, Romania; corneliu.toader@umfcd.ro (C.T.); bogdan.bratu@stud.umfcd.ro (B.-G.B.); luca-andrei.glavan0720@stud.umfcd.ro (L.A.G.); andrei.bordeianu@stud.umfcd.ro (A.B.); david-ioan.dumitrascu0720@stud.umfcd.ro (D.-I.D.); prof.avciurea@gmail.com (A.V.C.); 2Department of Vascular Neurosurgery, National Institute of Neurology and Neurovascular Diseases, 077160 Bucharest, Romania; 3Department of Opthamology, “Carol Davila” University of Medicine and Pharmacy, 020021 Bucharest, Romania; 4Central Military Emergency Hospital “Dr. Carol Davila”, 010825 Bucharest, Romania; 5Department of Neurosciences, “Iuliu Hatieganu” University of Medicine and Pharmacy, 400012 Cluj-Napoca, Romania; 6Neurosurgery Department, Sanador Clinical Hospital, 010991 Bucharest, Romania

**Keywords:** pitch perception, cognitive enhancement, memory encoding, limbic system, rehabilitation, music-based therapies

## Abstract

Music is a complex phenomenon with multiple brain areas and neural connections being implicated. Centuries ago, music was discovered as an efficient modality for psychological status enrichment and even for the treatment of multiple pathologies. Modern research investigations give a new avenue for music perception and the understanding of the underlying neurological mechanisms, using neuroimaging, especially magnetic resonance imaging. Multiple brain areas were depicted in the last decades as being of high value for music processing, and further analyses in the neuropsychology field uncover the implications in emotional and cognitive activities. Music listening improves cognitive functions such as memory, attention span, and behavioral augmentation. In rehabilitation, music-based therapies have a high rate of success for the treatment of depression and anxiety and even in neurological disorders such as regaining the body integrity after a stroke episode. Our review focused on the neurological and psychological implications of music, as well as presenting the significant clinical relevance of therapies using music.

## 1. Introduction

The inherent complexity of music renders it a multifaceted subject that eludes simple definitions. While many describe it as an ordered arrangement of sounds, musical elements such as harmony or the bass line require intricate understanding and considerable effort to master. In this research, our focus is on the neurological and psychological benefits of music listening, especially the potential usage of musical therapies, and how the brain might respond during varied activities set within a musical context [1].

Music is a universal phenomenon that utilizes a myriad of brain resources. Engaging with music is among the most cognitively demanding tasks a human can undergo, and it is identified across all cultures; therefore, it underscores its fundamental human nature [2]. The proclivity to create and appreciate music is ubiquitous among humans, permeating daily life across diverse societies [3]. This inherent connection to musical expression is deeply intertwined with human identity and experience. Molnar-Szakacs further emphasizes music’s unique capacity to evoke memories, stimulate emotions, and enrich social interactions [3]. Historical examples underscore the therapeutic potential of music. For instance, Johann Sebastian Bach’s Goldberg Variations (BWV 988) was purportedly composed to alleviate a count’s insomnia, underscoring music’s therapeutic potential [4,5,6]. The profound emotional impact of music, whether it be the melancholy evoked by a nocturne from F. Chopin or the elation induced by W. A. Mozart, has inspired ongoing research into its relationship with emotions and psychological disorders [7]. Fundamental to understanding music are the concepts of pitch perception, rhythm perception, and tonality perception.

### 1.1. Pitch Perception

Predominantly processed in the auditory cortex, pitch perception pertains to the brain’s handling of sound information. The auditory cortex features a tonotopic map wherein specific regions are sensitive to distinct frequencies. Human auditory perception ranges from 20 to 20,000 Hz, with distinct pitches resonating at precise locations on the basilar membrane. Yost et al. expound that understanding pitch necessitates a grasp of the biomechanical mechanisms and neurological shifts in sound as well as the diverse ways pitch can be conceptualized and potentially quantified [8]. Often, pitch is defined as the attribute of sound that sequences it from low to high levels. Musically, pitch aids in recognizing melodies and discerning intervals, with quantification methods ranging from equal-temperament tuning scales to the perceptive mel scale [9].

For instance, a standard 1000 Hz tone delivered at a 40 dB sound pressure level corresponds to 100 mels on the mel scale. It is important to note that variations in perceived pitch proportionately influence mel values. Much of pitch perception research delves into complex sounds, with the pitch of basic tones like sinusoids determined by frequency. Intricacies in encoding high-frequency and low-frequency tonal signals differentiate them, and while amplitude modulation is absent in simple tonal sounds, temporal mechanisms might play a role in low-frequency pitch perception [10].

In summary, understanding sound transformations, coupled with a range of definitions and measurement techniques, is imperative for accurate pitch perception. This encompasses melody recognition capacity, interval discernment, and frequency perception, with various mechanisms, both spectral and temporal, influencing pitch perception [11].

### 1.2. Rhythm Perception

Beat perception engages specific brain regions associated with motor planning and timing, notably the basal ganglia and the supplementary motor area. Interestingly, even passive listening to music can activate these neural domains [12]. The ability to discern a steady pulse underlying a rhythmic stimulus defines beat perception. This inherent pulse, which rhythmically structures the music, is an elemental consistency that the human cognitive apparatus innately detects. By accentuating beats in specific patterns, we can synchronize our movements (e.g., dancing or foot tapping) and regulate our temporal perception, culminating in the creation of meter. Rhythmic perception necessitates a combination of interval-based (absolute) timing and beat-based (relative) timing. While interval-based timing is observed in both humans and various animal species, beat-based timing might be unique to humans [13,14].

Motor theories centered on timing are primarily focused on beat-based timing. Active motor engagement seems to actively mold our perception of beats. For instance, the negative mean asynchrony effect, where one’s taps often precede the actual beat, underscores the pivotal role of anticipation in beat-based timing. Humans establish rhythmic timing anticipations and maintain a versatile perception of the intrinsic rhythmic architecture, even when confronted with alterations in tempo. Notably, rhythm perception is not merely passive; it is influenced by an individual’s active cognitive processing and volitional control, underpinned by metric interpretation [15]. Moreover, the very act of motor engagement shapes the perception of beats, manifests bodily movements, enhances temporal perception, and influences interpretations of ambiguous rhythms. Both overt motor actions and their covert counterparts play a role in refining perceptual sharpness. Even in scenarios devoid of visible motion, there is accumulating evidence that motor engagement modulates the perception of beat and meter. Contemporary research posits that the motor system not only influences beat perception but can also augment synchronicity with music [13]. Faster movements can also modulate the perceived pace of music segments [16].

To encapsulate, beat perception involves recognizing a steady pulse amidst rhythmic stimuli, a process that is dynamically shaped by motor activity, conscious modulation, adaptive tempo perception, and anticipatory mechanisms. Remarkably, even in scenarios devoid of overt motion, our sense of rhythm and meter remains intricately linked with the motor system [17,18].

### 1.3. Tonality Perception

The comprehension of key and harmony in music engages distinct neural domains, including the auditory, prefrontal, and parietal cortices. Scientific investigations are currently delving deeper into understanding the brain’s intricacies in processing musical harmony. The notion of harmony primarily stems from the amalgamation of sounds in Western tonal music. Within this musical paradigm, pitches are hierarchically arranged based on their congruence within a specific tonal context. Scales utilized in Western tonal compositions emanate from this pitch hierarchy. While the behavioral science community acknowledges the hierarchical essence of pitch organization, the neural substrates underpinning it remain a realm of exploration [19].

In a distinct study centered on J. S. Bach’s compositions, researchers probed the psychological relevance of musicians’ conception of tonality. Here, musically trained listeners were tasked with singing the first scale that resonated with them post hearing snippets from Bach’s Preludes in The Well-Tempered Clavier. The selected tonic (starting note) and mode (major/minor) were then juxtaposed against Bach’s original specifications. The data revealed that listeners could often discern the designated tonic and mode merely from the initial quartet of notes. However, as the piece progressed, there was a marked tendency to gravitate toward tonalities divergent from the original key, notably within the initial eight bars. By the concluding quartet of bars, the original tonic was often reaffirmed. Such findings not only spotlight the cognitive intricacies of tonality perception but also align with the postulations of music theorists regarding tonal discernment by listeners [20].

Tonality serves as the linchpin in music, underpinning the creation and comprehension of musical constructs such as melodies. A contemporary dynamic theory on musical tonality posits a nonlinear response of auditory neuron networks to musical stimuli. This tonal cognition, the intrinsic interconnections perceived amidst tones, arises from the robust and harmonious associations among brain frequencies, a phenomenon attributable to nonlinear resonance [21,22].

## 2. Materials and Methods 

We conducted a comprehensive search on PubMed database for the most relevant articles regarding music studies, musicology mechanisms, and music-based therapies. For the search formula, we used the following terms: “pitch perception”, “rhythm perception”, “tonality perception”, “memory encoding”, “limbic system”, “neuroplasticity”, “motor coordination”, “evoked memories”, “rehabilitation”, and “music-based therapies”. Initially, PubMed database showed 341 studies. Furthermore, each title of those articles was reviewed to include minimally one of the searching terms. Those studies that did not respect the inclusion criteria or were focused on other subjects besides musicology were excluded. After the analyses, only 132 studies were included in our study.

In this comprehensive review segment, we delve into existing studies, results, and theoretical postulations regarding the neurological implications of music and its therapeutic applications. The aim is to furnish a meticulous analysis of the current state of knowledge within this field, accentuating pivotal research endeavors, methodologies, and discoveries. Subdivisions within this section are delineated based on thematic content, research domains, or specific dimensions of the topic.

## 3. Results

### 3.1. Emotion and Reward Mechanisms in Musical Perception

Music possesses the unique capability to induce profound emotional responses, often intertwined with personal memories of significance. The neuroscientific underpinnings of this phenomenon suggest that music’s emotive power is rooted in the activation of the brain’s reward system. Notable neural regions involved include the nucleus accumbens and the ventromedial prefrontal cortex, elucidating the intrinsically rewarding and emotionally charged nature of musical experiences.

#### 3.1.1. The Interplay of Music with the Limbic System

Central to our emotional resonance with music is the limbic system, an intricate assembly of neural circuits and pathways. Key components of this system, such as the amygdala—responsible for emotional processing—and the hippocampus—integral to memory consolidation—become activated during musical exposure (Table 1). Such neural activities account for the evocative power of music to invoke vivid emotional and mnemonic experiences. The consequential effects can be observed when an individual is emotionally transported to a distinct temporal or spatial context upon hearing a particular musical piece or when a gamut of emotions is experienced in response to auditory stimuli [23].

During passive listening to unfamiliar yet positively perceived music, there was a spontaneous activation in both the limbic and paralimbic regions. Consistent with prior research on passive auditory experiences, primary and secondary auditory cortices displayed activations, corroborating findings from studies that analyzed listening to either monophonic or harmonized auditory sequences [24]. Furthermore, there were observed activations in the temporal pole, subcallosal cingulate gyrus, affective segment of the anterior cingulate cortex, retrosplenial cortex, hippocampus, anterior insula, and nucleus accumbens. It is plausible that these observed neuroanatomical patterns are a result of the intricate musical nature of the stimuli, which were highly favored by the participants. There is a prevailing theory suggesting that the left hemisphere predominantly facilitates positive emotions. This is in line with our findings that indicate a predominance of limbic and paralimbic activations on the left side, potentially mirroring the participants’ positive aesthetic reactions. The acquired functional neuroanatomical insights augment existing literature on music–emotion interplay, especially those employing high-temporal-resolution methodologies such as electroencephalography and magnetoencephalography [23].

Contrasting minor with major melodies showed multiple activation sites (Figure 1) with the right parahippocampal gyrus (RPHG) being an eloquent brain area (Figure 2). Another discernible activation, when subjected to cluster-level correction, spanned both the left and right ventral anterior cingulate cortex (VACC) (BA 24) and extended into the left medial frontal gyrus (LMFG) within the medial prefrontal cortex (BA 10) (Figure 3). Remarkably, the inverse contrast (major over minor) did not yield significant activations. In a peak-voxel analysis, the response to the chromatic scale was intermediary when juxtaposed against the major and minor mode melodies for three of the aforementioned regions. These differential responses between the chromatic scale and melodies were not statistically significant, with an exception. Within the LMFG, the chromatic scale evoked the most prominent (least negative) response, trailed by the minor and subsequently the major mode. Notably, the contrast between the chromatic scale and the major mode was statistically significant in this context [25].

VACC activation is generally associated with affective processing, while its dorsal counterpart is linked with cognitive functions [26]. Moreover, the existing literature indicates that the VACC displays heightened sensitivity to emotional content characterized by negativity or sadness [27]. The observed engagement of the VACC might be consistent with the perception of minor mode melodies as possessing a sadder tonality in comparison to major melodies. Notably, prior neuroimaging research on mode-based contrasts has not reported VACC activation in contrasts between minor and major modes [28].

The detected involvement of the left medial frontal gyrus (LMFG) may be attributed to its robust neural connectivity with the anterior cingulate cortex and other limbic systems. Such a connectivity profile underscores the proposed function of the medial prefrontal cortex as an integrative nexus for emotional input from these associated regions [29].

Research encompassing neuropsychology, neurophysiology, and health science domains suggests that patients in a low-awareness state exhibit both anatomical and behavioral divergences in response to auditory stimuli. These differences underline the auditory channel’s pivotal role in evaluating such patients. More specifically, the distinct auditory responses between individuals in a vegetative state (VS) and those in a minimally conscious state (MCS) when exposed to emotionally significant auditory stimuli imply that interventions incorporating personally resonant auditory content could lead to discernible outcomes, thus aiding diagnosis. However, diagnostic endeavors are often confounded by non-intentional emotional, or “limbic”, reactions observed in VS patients [30].

Multiple studies have documented elevated neural activity in MCS patients when exposed to emotionally significant auditory cues, suggesting these individuals possess the capability for discriminatory auditory responses. For instance, Boly et al. observed that stimuli like distress calls or a patient’s own name elicited more extensive neural activations compared to irrelevant noises [31]. In addition, cognitive-evoked potentials in response to an individual’s own name differed from those induced by other names, reinforcing the clinical premise and observational data that personally significant stimuli are more likely to induce pronounced behavioral alterations [32].

#### 3.1.2. Music Seems to Encourage Enhanced Connectivity between the Auditory and Emotional Regions of the Brain

Listening to music engages not only the auditory cortex, responsible for sound processing but also several emotional centers within the brain. For instance, a musical composition perceived as melancholic might enhance the connectivity between the auditory cortex and the hippocampus, a region integral to memory and emotional processing. This interconnection can trigger the recollection of somber memories or evoke feelings of sadness.

Activations were prominently observed bilaterally in the anterior sections of the middle and superior temporal gyri. Prior research has identified the anterior temporal lobe’s involvement in comprehension at the sentence level, distinct from the temporal assimilation of significant auditory cues. Notably, this region’s activation is rather selective for sentence-level stimuli. It does not exhibit pronounced responses to unstructured meaningful auditory cues like word lists or random sequences of environmental noises. Nevertheless, it does react to both coherent sentences and nonsensical pseudoword sentences. The study’s authors noted the unresolved question of whether this region also becomes active during musical engagements [33].

Positron emission tomography (PET) studies focused on auditory imagery for music have documented the active involvement of the supplementary motor areas (SMAs) during image generation. This indicates the SMA’s potential role in an internalized “singing” process during auditory imagery tasks [34,35]. However, these studies did not explicitly associate SMA activity with the rhythmic elements of music. Notably, research involving patients with SMA lesions unequivocally demonstrates their difficulties in replicating rhythms [36]. The observed diminishing correlation of SMA activity with rhythmical performance following each alteration in the degree of temporal deviations from the reference interval ratio (DRIR) mirrors the decline in SMA activity as a motor task is reiterated. This parallel highlights the analogous motor-related neural activations during both motor activities and musical perception [37].

### 3.2. Motor Systems

Engaging in musical activities necessitates intricate motor tasks that demand precise timing and coordination. The cerebellum, an integral part of the brain dedicated to timing and motor coordination, demonstrates heightened activity among musicians. Other motor-related regions, such as the premotor cortex and the basal ganglia, play pivotal roles in both producing and perceiving music. Comprehensive motor systems, spanning from fine motor skills to broad motor coordination, are crucial for regulating the physical actions inherent in playing a musical instrument or singing [38].

Fine Motor Control: Precision in playing musical instruments necessitates exceptional motor control, specifically in muscles such as the fingers and hands.

Finger Dexterity: Musicians cultivate nuanced finger motions, granting them the capability to adeptly handle their instrument’s keys, strings, or frets. This proficiency enables diverse pitch generation and the execution of intricate melodies or chords. Notably, pianists, aspiring to master compositions like Liszt’s “Transcendental Studies”, S. 139, or Beethoven’s Piano Sonata No. 21 “Waldstein”, Op. 53, commonly practice upward of 6 h daily [39].

Hand Coordination: Instruments such as pianos or guitars necessitate meticulous coordination between hands. A harmonious interplay is required where one hand typically manages the melody or leads, while the counterpart offers harmonic or rhythmic accompaniment [40].

Embouchure Control: Wind instrument performers, encompassing flutists and trumpeters, are reliant on meticulous muscle control of their mouth and lips for tone production and airflow modulation [41].

Gross Motor Coordination: Distinct from precision-centered fine motor control, gross motor coordination emphasizes the integration of larger muscle group activities.

Body Movement: Many musicians incorporate physical gestures to accentuate rhythm or enhance their presentations, such as rhythmically swaying or foot tapping [42].

Posture and Breathing: Vocalists and wind instrument practitioners stress the importance of appropriate posture and breath management. Optimal posture underpins efficient breathing, ensuring voice projection and breath modulation [43].

Sensorimotor Integration: An intimate synergy between motor coordination and sensory feedback is paramount for musical endeavors.

Visual Feedback: Musicians harness visual indicators like music notations or the synchronized actions of co-performers to facilitate timing coordination and group harmonization [44].

Tactile Feedback: Musicians depend on tactile sensations and muscle memory, underpinning finger positioning and pressure modulation on their instruments [45].

Auditory Feedback: By closely monitoring their auditory output, musicians can fine-tune pitch, pace, and tonal quality. This auditory feedback loop enables real-time adjustments, promoting accuracy [46].

In summation, the intricate interplay of fine motor skills, gross motor coordination, and sensorimotor integration embodies the complexity of musical performance. Through relentless training and practice, musicians refine their motor capabilities, striving for both mastery and evocative expression.

#### 3.2.1. Music and Rhythm Processing

Music, at its core, engages our motor systems predominantly through the element of rhythm. The basal ganglia and the supplementary motor area (SMA) stand out as pivotal neural regions governing rhythm processing. Specifically, the basal ganglia take center stage in organizing movements, determining timing and sequencing, and forecasting forthcoming rhythmic beats [47,48].

To delve deeper into rhythm cognition within music, one should familiarize oneself with the PRISM framework. This framework elucidates three central mechanisms: precise auditory processing, synchronization of brain oscillations to rhythmic stimuli, and the interplay between sensory perception and motor action known as sensorimotor coupling. Collectively, these mechanisms facilitate rhythm processing in both musical and speech domains [48,49].

Accurate Auditory Processing: This entails discerning minute time deviations and provides the bedrock for rhythm perception, enabling the detection of intricate temporal patterns.

Brain Oscillation Synchronization: This mechanism concerns the brain’s ability to anticipate ensuing events and conform to hierarchical rhythm structures. It ensures the alignment of rhythmic components, contributing to the holistic rhythm experience [47].

Sensorimotor Coupling: This establishes a link between perception and execution, implicating the motor system in tasks like timing, prediction, and integrating auditory cues with motor actions.

The PRISM framework offers an innovative lens through which rhythm processing in music and speech is perceived. By illuminating shared neural mechanisms between music and speech, this model enriches our understanding of rhythm processing, thereby opening avenues for further research, particularly in the arena of speech and language impediments [48].

However, beyond the neurocognitive realm, rhythm perception and production are intertwined with cultural nuances. While cognitive and physiological components might offer a universal rhythm perception baseline, cultural experiences undeniably play a significant role. Infants, for instance, exhibit an inclination toward rhythmic patterns emblematic of their culture’s music, suggesting cultural influences even at infancy [50]. Cultural aspects also influence language rhythm perception, with speech patterns often mirroring a given culture’s musical rhythms. A comparison between Western and East African music presents a stark difference in rhythm complexities and significance, highlighting the cultural diversity in rhythm processing [51].

Furthermore, cultural disparities might not only dictate how rhythm is perceived but also the range of rhythmic frequencies one aligns with. For instance, African music’s inherent metrical ambiguity might afford listeners the flexibility to engage with multiple rhythmic levels, diverging from the more rigid Western musical counterparts [52].

In summary, rhythm’s multifaceted nature intertwines neural processing with cultural nuances. Cultural exposure and familiarity undoubtedly mold our rhythmic preferences and processing capabilities, underscoring the intricate relationship binding music, language, and societal constructs [14].

#### 3.2.2. Music and Motor Coordination

Playing a musical instrument, especially the piano, is a testament to the intricate dance of our motor systems. Brain scans of musicians highlight heightened activity in regions like the motor cortex and cerebellum, both critical for motion and coordination. Notably, the cerebellum emerges as the linchpin for fine motor control and timing, skills indispensable to instrumentalists. Music is not just an art; it reshapes the brain. Lifelong musical tutelage can cause an enlarged motor cortex and cerebellum, imprinting physical markers of musical expertise [53].

One striking feature of our motor system is its redundancy. With a plethora of joints and muscles at our disposal, multiple movement combinations can yield the same outcome. Renowned pianists like Martha Argerich and Dinu Lipatti exemplify this by leveraging redundancy to achieve specific acoustic effects, each using unique motor configurations [54]. This fluidity arises from neuroplasticity, where the neuromuscular system continually reshapes itself, enhancing the finesse of advanced motor activities. By juxtaposing skilled versus novice pianists, researchers probe into the interplay of neuroplasticity, motor redundancy, and the nuanced organization of piano-playing movements. While gauging the long-term impact of training remains challenging, such comparisons offer valuable glimpses into the artistry of motor skills [55].

The redundancy in pianists’ motor systems is multilayered. They can achieve the same note with various force and movement patterns at the fingertip, navigate multiple joint combinations to produce identical fingertip movements, and leverage various forces to generate the same joint rotation [56]. Amidst this intricate web, muscular torque stands out. It is birthed from the balance of forces exerted by opposing muscles around a joint. Given the motor system’s richness, pianists have countless ways to strike a single note. Masters of the craft excel in navigating this maze by optimizing energy use, achieving physiological efficiency. Their prowess is evident in their enhanced coordination, minimal muscle discomfort, and adeptness at offsetting mechanical interactions [50].

How pianists employ joint rotations and balance various forces exemplifies the interplay of kinematic and kinetic configurations. Elite pianists adopt strategies like optimized postures and sequential joint movements, optimizing movement and conserving muscle energy. By harnessing gravity, they also conserve energy when pressing keys, further showcasing motor redundancy. A consistent finding in studies contrasting expert versus novice pianists is the former’s unique upper limb motion organization, honed through rigorous practice. Such an organization is attuned to physiological efficiency, minimizing energy costs for known tasks. It is no surprise then that seasoned pianists, even in demanding performances, manage to retain their performance quality, all while fending off muscle fatigue [57].

In a fascinating dive into the world of jazz improvisation, Setzler M and colleagues explore how mutual coupling influences the coordination dynamics of professional jazz performers. The study revolves around understanding the interplay of rhythmic and tonal patterns as musicians exchange and spontaneously produce musical elements. With expert pianists from the vibrant New York City jazz circuit as participants, the study juxtaposes a unique one-way scenario, where a pianist improvises to a pre-recorded duet, against two dynamic duo conditions: a coupled setting where both pianists are improvising in real-time. While the one-way setup showcases unilateral coordination, in the duo scenario, the pianists adjust to each other’s rhythms and tones. The catch? The improvisations are uninhibited by any predefined song structure, key, or tempo [58].

The study dives deep into the data, examining parameters like tonal consonance (how harmoniously musical combinations sound) and onset density (the extent of rhythmic activity). The findings are illuminating: when pianists are connected and responding to each other in the duo setup, they consistently exhibit enhanced coordinated behavior. They create more harmonious tonal structures and display heightened rhythmic synchronization, compared to the unilateral one-way condition. Notably, these observations align with both the pianists’ personal experiences and the auditory preferences of lay listeners [58].

But why does this matter? The implications of this research are manifold. Firstly, it propels the domain of collaborative action studies and music technology. By understanding the nuances of how mutual coupling impacts musical coordination, we gain insights into complex, unrestrained coordination typical of stellar artistic performances. Such insights go beyond controlled lab environments. Moreover, the findings can shape the future of interactive music systems, potentially revolutionizing how ensemble performances are evaluated in musical training. The study’s roster boasts 28 seasoned pianists, all with robust backgrounds in jazz improvisation, along with a diverse listening panel comprising both jazz maestros and undergraduate psychology students. In essence, this research provides a valuable lens into the intricate dance of coordination during musical improvisation, shedding light on how it elevates the quality of the resultant melodies [59].

#### 3.2.3. Music and Rehabilitation

Music’s healing touch has progressively found its way into motor rehabilitation, offering a rhythmic respite to those grappling with motor skill challenges. Music-based therapeutic interventions, for instance, have emerged as powerful tools for stroke patients, helping them regain lost motor functions. The rhythmic predictability embedded within music seems to have a harmonious effect on patients with Parkinson’s disease, addressing their movement-related issues like gait and timing disruptions. This rhythmic auditory stimulation (RAS), as is known, offers an external rhythmic pulse that works wonders in steadying and regulating motor timing. This incorporation of music in treating age-related neurological ailments is backed by numerous studies [6].

The global surge in age-related neurological disorders, propelled by an aging population, has escalated the economic burdens associated mainly with non-acute treatments. This has ignited the quest for cost-efficient rehabilitative methods to complement traditional approaches like physiotherapy. While there is a limit to how much adult brain neurogenesis can contribute to healing, functional restoration does not share this limitation. Shifting from targeted training of impaired functions, some modern methods are championing a holistic rise in brain activity through sensory and cognitive stimulations [4].

Research has illuminated how musical pursuits like playing an instrument can reshape the brain. Even mere listening to music has been observed to bolster neuronal connections in certain brain areas, such as the auditory and visual cortices. Music’s therapeutic touch extends to post-operative recovery as well, alleviating pain and anxiety and reducing the dependence on painkillers [60]. Certified music therapists employ both active and receptive music-based therapies, encompassing musical expressions ranging from singing to playing instruments [61]. While initial studies revolved around music’s impact on acquired brain injuries, comprehensive investigations into its effect on major neurological diseases are still unfolding [6].

The review delves into music-based therapies’ impact on ailments like stroke, dementia, Parkinson’s, epilepsy, and multiple sclerosis, gauging the therapies’ efficacy through randomized controlled trials. The “effect size” metric offers insights into the degree of improvement observed [5].

Further, the study zooms in on the potential of dance and RAS in rehabilitating individuals with cerebral palsy (CP) [62]. Preliminary evidence champions the benefits of dance and RAS in enhancing physical functionalities, especially areas like balance, walking, and cardiorespiratory fitness in CP patients. Despite the extensive categories in the International Classification of Functioning, Disability and Health (ICF), there remain research voids, especially in areas concerning participation and environmental factors [63]. Bridging these gaps, the review synthesizes quantitative rehabilitation findings within the ICF framework, pinpointing further research avenues. It concludes by celebrating dance and RAS’s potential in enhancing not just physical processes but also emotional expression, social interactions, and overall well-being [64].

#### 3.2.4. Entrainment

Entrainment, a captivating phenomenon where we unconsciously synchronize our movements to an external rhythm, emerges as an inherent human response when engaged with music. This almost involuntary response—be it foot tapping or dancing—is not just about moving to the beat. It is an intricate interplay of various brain regions responsible for auditory processing, motor functions, and even prediction [65].

Music, a rich tapestry of sensory, cognitive, and emotional experiences, is not just about the melody or rhythm. When we engage with music, it evokes a spectrum of emotions—from joy and sorrow to more nuanced feelings like wonder or nostalgia. These complex emotions do not necessarily fit into conventional neuroscientific emotion categories, leaving a vast realm still largely unexplored [66]. The authors delve deep into these intricate emotions, suggesting they are possibly birthed from the confluence of multiple brain areas, including those responsible for attention, motor functions, and memory, intertwined with emotional and motivational pathways. Such an understanding holds profound implications, especially in therapeutic realms, potentially aiding conditions marred by attention, motor, or affective disruptions [67,68].

“New Music” presents another dimension to our musical discourse. Unlike its classical counterpart, defining “New Music” is like capturing lightning in a bottle—it is ever-evolving, challenging norms, and shunning traditional tonality and rhythms. The listener, when immersed in the world of New Music, must recalibrate their cognitive tools to truly appreciate this avant-garde genre. While it is a mosaic of styles, some dominant shades include the second Vienna School, electronic synthesis, microtonal music, and more [9,69]. Branching further, genres like Ambient Music and Postclassical Minimal emerge, each with its unique essence.

Recognizing the need for a deeper dive into “New Music” and its neurological interplay, a dedicated research topic was launched, casting a wide net from embodied cognition to technological impacts and even neuroimaging techniques like EEG and fMRI [9]. The selected studies ventured into diverse terrains—from tempo perceptions, the philosophy of sound objects, and networked music performances to the nuances of atonal music, especially with pioneers like Arnold Schönberg at its helm [70]. Truly appreciating New Music mandates unconventional cognitive frameworks, from embodiment to heightened attention to recurring or absent elements. Functional brain imaging, though still in its nascent stages, promises insights into our cerebral engagement with these novel musical narratives. While the current discourse sheds light on New Music’s mysteries, a harmonious symphony of extensive and collaborative research is imperative for a deeper understanding [71].

### 3.3. Memory

Music and memory share an intimate bond. Often, a song can trigger a cascade of vivid memories, while melodies and lyrics, even from years past, can be effortlessly recalled. Such connections correlate with activations in areas like the hippocampus, pivotal in memory storage and retrieval.

#### 3.3.1. Memory Encoding with Music

Harnessing a song’s melody and rhythm can be a powerful mnemonic device. Information set to a catchy tune tends to stick, an approach adopted in education to teach topics ranging from languages to science.

Smith et al. (1985) posited a compelling idea—using music as a backdrop during the encoding of words can be a catalyst in context-dependent memory during retrieval. This effectively boosts the recall of the encoded words [72]. Extending this thought, there is mounting evidence that suggests music’s potency in facilitating episodic encoding of events [73]. Across various studies, employing musical stimuli like background tunes or sung texts consistently showed improvements in verbal memory for both standard [74] and clinical groups [75,76]. However, while these studies underscore music’s ability to enhance the recall of encoded items, most have not delved into the musical context during the retrieval process. Among those that did, outcomes have been mixed [77].

Using fNIRS studies, it has been found that musical backdrops during verbal material encoding can bolster both item and source memory, linked to the modulation of prefrontal cortex activity [78]. However, some limitations exist, primarily since these studies only compared musical contexts to silence, leaving unanswered questions regarding the impact of non-musical auditory stimulations on memory.

Contrary to the majority, El Haj et al. (2014) proposed that musical backgrounds might impede source memory performance across age groups, adding more layers to the ongoing debate [77].

Ferreri et al. (2015) shed more light on the subject, indicating that specifically a musical backdrop (and not just any sound) can enhance verbal encoding. The ongoing discussion shifts to which specific elements of music augment memory. Past research has indicated that factors like perceptual characteristics, the emotional undertone, and interpretive variations in musical stimuli play pivotal roles in boosting memory and learning [79]. Adding depth to this understanding, it is noted that emotional inputs modulate musical memory akin to their influence in other domains [80].

A salient aspect of the music–memory nexus is the role of rewarding stimuli in cognitive tasks. Music stands tall as one of the most rewarding stimuli, and recent insights suggest its potential in augmenting cognitive performance [81].

#### 3.3.2. Evoking Memories

Music has an uncanny ability to immerse us back into past moments, often reviving the very emotions we felt during those times. This phenomenon arises because music not only captures the essence of our emotional state when memories form but also acts as a potent cue to rekindle them. Thus, a mere tune or lyric can instantaneously propel us to a distinct time or place, evoking associated feelings.

This intricate bond between music and memory has led to the term “musical memory”. This refers to the unique connection between certain songs and personal experiences, elucidating why particular melodies can instantaneously remind us of specific past events or people.

Music-evoked autobiographical memories (MEAMs) are often charged with intense emotions—be it joy, excitement, or nostalgia [82]. For instance, a study by Janata et al. (2007) found that popular music-triggered MEAMs were profoundly emotional. They noted that when participants resonated deeply with a song, they were more inclined to associate it with a personal memory [83]. Neuroimaging research supports the emotionally charged nature of MEAMs and illuminates music’s ability to evoke memories of varying specificity [84]. Such revelations underscore music’s prominence as a memory catalyst.

Research also explores music’s role in memory recall among Alzheimer’s patients. For instance, Foster and Valentine (2001) noted that Alzheimer’s patients retrieved more personal memories post music exposure compared to when exposed to white noise or silence [85]. Similarly, a study by Irish et al. (2006) found that Alzheimer’s patients exhibited enhanced episodic memory recall when exposed to Vivaldi’s Spring from the Four Seasons [86]. However, since music in these studies played in the background and not as a direct memory cue, the results showcase music’s influence on memory recall but do not differentiate music-evoked memories from those elicited by other stimuli.

When pitting memories triggered by music against those by faces, the former emerged as more vivid. However, the total number of internal details remained consistent across both. The primary distinction was in external details—face-induced memories contained more such details, often rich in semantic information about the pictured individual [87]. Interestingly, gender dynamics were evident in memory retrieval; women consistently described more vivid autobiographical memories than men, regardless of the cue. Several studies have hypothesized that this could be attributed to gender-specific encoding styles, with women registering memories more intricately. Additionally, Piefke et al. (2005) proposed that men and women employ distinct cognitive strategies during memory retrieval [88]. Another variable impacting the vividness of autobiographical memories is age. Typically, older adults recall memories that are less specific and contain fewer episodic details compared to their younger counterparts [89].

#### 3.3.3. Neurological Basis

Our understanding of music’s influence on the brain is intricate, involving numerous regions that process auditory information, emotions, and memories.

During music perception, the auditory cortex plays a central role, processing the sound. Simultaneously, areas associated with emotional responses, like the amygdala, and memory, such as the hippocampus, become activated. The medial prefrontal cortex is particularly interesting; it springs into action when we hear familiar tunes. It is also significant to note that this region is one of the last to degenerate in Alzheimer’s disease, hinting at its role in the robust link between music and autobiographical memories.

Delving into the neural mechanics of music performance, Langheim et al. (2002) discovered activations in various brain areas, including the supplementary motor and premotor regions, right superior parietal lobule, right inferior frontal gyrus, bilateral midfrontal gyri, and the bilateral lateral cerebellum, during imagined musical performance. Notably, they did not observe activation in primary sensorimotor areas and auditory cortices [90]. The specific activation of the right inferior frontal gyrus is believed to be tied to music production. This idea is reinforced by other studies highlighting the involvement of this area in selective attention, working memory, and motor synchronization with auditory cues [91,92].

In another exploration, Nirkko et al. (2001) demonstrated that playing a musical sequence on a violin led to activation in several brain regions. Notably, they highlighted the involvement of bilateral fronto-opercular regions, suggesting their role in timed motor sequences present in both music and language production [93].

Another crucial region, the superior temporal gyrus, processes complex patterns formed by individual musical notes [94]. Platel et al. (1997) observed that the activation of specific parts of this gyrus, coupled with the left inferior frontal gyrus, indicates semantic access to melodic elements [95].

Popescu et al. (2004) noted early activations around primary and secondary auditory cortices, as well as in posterior parietal areas post stimulus onset [96]. These regions are critical for language and music processing. Furthermore, activations in the supramarginal and postcentral gyri have been associated with processing the basic attributes of sound [97]. Meanwhile, music listening’s impact on the precuneus has been documented in several studies, emphasizing its role in sound processing [98].

In sum, our brain’s reaction to music, whether in perception or production, is a symphony of neural activations across multiple regions, underpinning our rich emotional and cognitive experiences with melodies.

#### 3.3.4. Music in Therapeutics

Music, renowned for its potent link with memory, has been harnessed therapeutically in numerous medical scenarios. In conditions like Alzheimer’s disease and dementia, where individuals frequently grapple with short-term memory loss, familiar tunes can rekindle past memories and experiences. This often enhances mood and bolsters social interactions. In the realm of stroke rehabilitation, music therapy has been instrumental in assisting patients in regaining verbal memory (Figure 4).

The potential of music-based interventions in the neurological realm is immense, particularly in mending motor or cognitive functions. However, the design of these interventions often targets a specific pathology group. Among these, the evidence is most compelling for bolstering motor skills in stroke patients. It is imperative to approach these findings with caution; there is a risk of attributing improvements solely to the interventions, overlooking the role of natural recovery. Some studies, for instance, that employed bimanual piano training or gait training to musical cues, may not have utilized the most accurate measures to gauge improvements in coordination, dexterity, or balance. Nevertheless, the adaptability of music-based interventions in a clinical setting is noteworthy. They can be tailored to the individual, offering both a progression in therapy and personalization in treatment choice [99].

One salient area where music-based interventions shine is in addressing cognitive–motor interference, a common challenge in many neurological ailments [100]. Parallel executive deficits can sometimes hinder the effective rehabilitation of cognitive or motor shortcomings [101].

Here, music-based approaches emerge as dual-task training, transcending mere motor or cognitive training. Consider, for instance, an intervention utilizing a musical instrument. Here, the act of producing music, which involves moving parts of the body like the fingers (motor system), dovetails with the cognitive system, which processes new musical data, such as rhythm or pitch. This is especially pertinent, as significant cognitive–motor disruptions arising from such dual tasks are prevalent in many neurological conditions and can escalate the risk of falls [102,103].

**Figure 4 brainsci-13-01390-f004:**
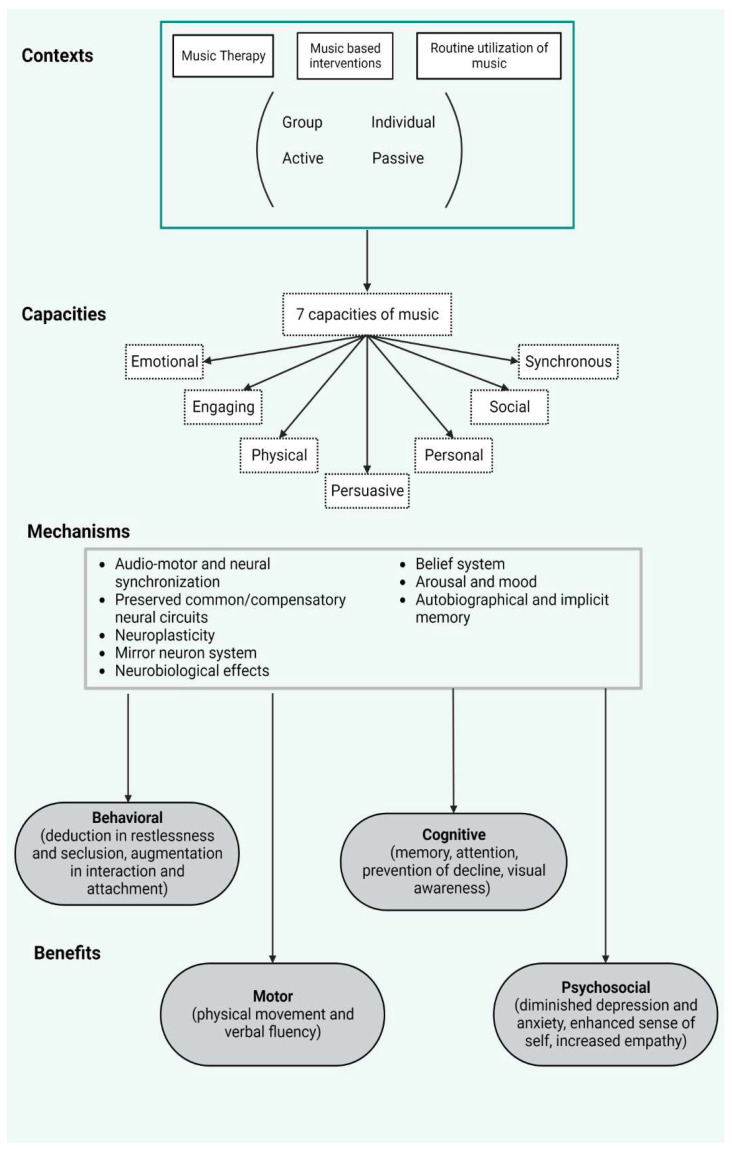
This image depicts examples of the main possibilities of clinical therapies using music. The given context is music therapies and daily music listening in various situations, such as in groups or individually and active or passive listening. Music offers multiple cognitive advantages and might be perceived in multiple ways which are described as “capacities”. Underlying mechanisms of music processing were aforementioned in this study, audio-motor functions and neuroplasticity being of high interest. Multiple behavioral-cognitive benefits, as well as motricity and psychological status, are highly improved [61,104]. Preprinted from Brancatisano, Baird, and Thompson, 2020 [61], with permission from the authors.

In essence, the therapeutic power of music, weaving together cognitive and motor systems, can be a beacon of hope in the multifaceted landscape of neurological rehabilitation.

### 3.4. Language Processing

Music and language are intricately linked, both weaving together structured sequences of sound and resonating in similar domains of the brain’s left hemisphere. Evidence suggesting that musical training can bolster language skills lends weight to the idea that the neural mechanisms underpinning both might overlap.

#### 3.4.1. Musical Training’s Influence on Language Skills

It has been established that musical education can enrich language abilities. This is likely due to the shared demands both disciplines place on discerning differences in pitch, timing, and tone. Musicians often exhibit heightened skills in phonetic discrimination (the capacity to differentiate between speech sounds), enhanced verbal memory, and superior reading capabilities. Furthermore, rhythmic competencies, which are sharpened through music, correlate with improved reading and linguistic prowess.

The nexus between musical training and intelligence has been a subject of rigorous debate in recent times. Music training and its duration have been consistently linked with higher intelligence across age groups, from children to adults [105,106]. This relationship is evident in various studies, where notable differences in non-verbal reasoning skills emerge between those with and without musical training [107,108]. Moreover, a correlation exists between non-verbal intelligence and musical aptitude. However, a potential confounder is the fact that children who receive music lessons often hail from affluent backgrounds, which could potentially skew the interpretation of these findings, especially in studies relying on correlation [109,110].

Recent data reveal that consistent participation in music playschools augments the development of phoneme processing abilities and vocabulary in children aged 5–6. In contrast, dance lessons did not exhibit a comparable impact. The disparities in children’s development crystallized over our two-year monitoring period. Interestingly, children exposed to both musical and dance education did not display a noticeable edge in vocabulary development. One theory is that these children had relatively high scores at the onset, leading to a potential ceiling effect as the study progressed. However, by the study’s conclusion, children who only attended music playschool and initially exhibited lower scores gravitated toward the higher-scoring group. This suggests that music-centric activities might especially benefit children who initially lag in linguistic tasks, at least within the observed age bracket of 5–6 years [111].

In essence, the confluence of music and language is undeniable, and the enriching impact of musical training on linguistic skills is evident, shedding light on the profound interconnectedness of these domains.

#### 3.4.2. Music and Speech Prosody

Music’s ties to the prosody of speech are compelling. Prosody, encompassing pitch, rhythm, and volume, is pivotal for embedding emotion and context in speech. Notably, these facets are fundamental to music, underlining a profound link between the musical and the expressive elements of language.

However, the waters are murkier when exploring pitch. While studies on pitch perception distinctly delineate between global and local processing, research on human voice recognition often treats pitch as a unified acoustic/perceptual element. The role of pitch in identifying talkers remains an enigma. Idiosyncratic prosodic alterations, especially the dynamics of the F0 contour, prove useful for distinguishing speakers [112]. However, absolute pitch height is another identifier, rooted in the individual’s unique laryngeal structure. For instance, by adjusting the pitch of synthetic speech, one can shift listeners’ perception of the number of dialogue participants [113]. A focus on individual differences can illuminate pitch perception’s role in talker identification, potentially disentangling the different ways pitch is processed in this context. Earlier studies indicate that global and local pitch processing can be separated, especially when linked to other linguistic proficiencies like reading [114]. Connecting differences in global versus local pitch perception with listeners’ variability in talker identification can provide a clearer understanding of pitch’s role in this process.

Long-term musical training is reputed to enhance pitch discernment [115]. This perceptual edge extends beyond musical pitch, touching the linguistic realm. A compelling connection between music and language is evident in studies examining lexical tone processing. For instance, musical training or aptitude can predict non-tonal language speakers’ prowess in identifying lexical tones [116] and in mimicking them, as well as their competency in learning them [117].

These findings, which underscore improved talker identification via experience in music and language, carry significant ramifications for understanding auditory perception’s adaptability. The evidence suggests that long-term engagement with music or consistent lexical tone use can augment listeners’ pitch sensitivity. This challenges the rigid compartmentalization of cognitive systems dedicated to music, language, and talker identification, pointing toward a more fluid, interconnected cognitive landscape [118]. In essence, music and language are not just standalone entities but intertwined realms, each enriching the other.

#### 3.4.3. Therapeutic Applications

The bond between music and language not only provides insight into cognitive function but has also been harnessed for therapeutic means. A salient example is melodic intonation therapy (Table 2), a method designed to assist aphasia patients (those who have lost language abilities typically due to brain damage, often resulting from a stroke) in regaining their speech. By engaging a patient’s preserved musical processing abilities, the therapy facilitates language recovery.

However, the therapeutic application of this connection has seen a myriad of interpretations. Initial accounts [119] present deviations from the original protocol, indicating the use of three pitches instead of the initially outlined two. Anecdotal evidence further showcases this diversity: therapists, based on observational data from across the U.S., each bring their own flair to the technique. Variations range from employing two pitches with specific intervals and crafting unique melodies for phrases incorporating multiple pitches to using the piano as an accompaniment or even tapping a sequence of notes on a patient’s arm as words or phrases are sung. Such diverse interpretations, while possibly tapping into right hemisphere regions pivotal for speech, might deter therapists lacking a musical foundation from adopting the therapy, given its intricate nature [120].

Additionally, the act of tapping the left hand could activate the right hemisphere’s sensorimotor network, responsible for both hand and mouth movements [121]. This action might bolster sound–motor mapping—an essential facet of meaningful vocal exchanges. Moreover, akin to the consistent beat of a metronome, tapping could offer a rhythmic guide, ensuring regular pacing and ongoing cues for the production of syllables [122]. In essence, the nuanced interplay between music and language has profound therapeutic potential, albeit varied in its execution.

## 4. Discussion

Music training has been identified as a catalyst for neurological transformation, exemplifying the phenomenon of neuroplasticity. Notably, individuals with a background in music often exhibit more pronounced auditory and motor regions compared to their non-musical counterparts. Such changes have far-reaching implications, encompassing areas like memory enhancement and heightened attention [123].

### 4.1. Anatomical Adaptations

Long-term involvement in musical endeavors can result in discernible anatomical shifts within the brain. Such transformations mirror the refined skills inherent to musicians, encompassing areas like auditory discernment, sound-associated emotional interpretation, and intricate motor control. Musicians, for instance, typically possess a more substantial corpus callosum—the neural bridge uniting the brain’s two halves. This could arise from the necessity of synchronized hand movements or the amalgamation of sensory–motor data. Moreover, regions governing motor functions, like the precentral gyrus, often exhibit greater development in musicians. Likewise, areas pivotal for auditory functions, such as the superior temporal gyrus, are frequently more evolved [124].

### 4.2. Operational Modifications

Beyond anatomical alterations, enduring musical training can usher in functional adaptations. When undertaking specific tasks, musicians often demonstrate unique brain activation patterns, emphasizing the brain’s adaptability in response to persistent training. For instance, the auditory cortex in musicians may exhibit heightened activity during music perception, signifying their adeptness in deconstructing musical elements [125].

### 4.3. Neurochemical Interactions and Neuronal Growth

Musical interactions influence more than just the brain’s physical contours; they also modulate its internal chemistry. Engaging with musical elements can spur dopamine release, linked with pleasure sensations, serotonin, regulating mood, and oxytocin, associated with social trust and bonding [126]. Furthermore, music might bolster neurogenesis, or the genesis of novel neurons. Preliminary animal research suggests that music exposure can amplify hippocampal neurogenesis, a core component in learning and memory. While promising, particularly concerning conditions like Alzheimer’s, further studies are imperative for a comprehensive grasp [127].

### 4.4. Cognitive Enhancement through Music

Music-induced neuroplasticity can elevate cognitive prowess, transcending just musical abilities. Musical children often outpace their non-musical peers in areas like reading, linguistics, and mathematical proficiency. Additionally, their attention span, memory, and executive functionality are frequently more advanced. Such augmentations are theorized to emerge from the transfer effect, where proficiency in one domain (e.g., music) amplifies skills in another (e.g., math). In essence, the cognitive tools sharpened by musical immersion—such as pattern detection and motor coordination—might be applicable across diverse domains [128].

### 4.5. Therapeutic Application of Music

The neurological adaptability influenced by music has been harnessed therapeutically, especially in neurorehabilitation post traumatic events like strokes [129]. Music-centric therapies can instigate restorative neuroplasticity. An illustration is music-supported therapy, wherein patients rehabilitate motor functions by playing musical instruments. Playing instruments mandates recurrent, meticulous movements, essential for reinstating motor command. Furthermore, the intrinsic reward of music amplifies patient motivation [130]. Another intervention, melodic intonation therapy (MIT), targets non-fluent aphasia patients, aiding their speech recovery. This method capitalizes on the brain’s adaptive potential, utilizing unharmed singing capacities to reinvigorate linguistic prowess [131].

An important point of view for an efficient rehabilitation process is using comprehensive approaches, especially in those patients who suffered a myocardial infarction or an ischemic stroke. In a recent study [132] focused on the effect of implementing robot-assisted physiotherapy technology for heart infarction treatment, great results were obtained in ADLs (activities of daily living) and motor functions. Moreover, in ischemic stroke scenarios, multidisciplinary combined healthcare management provides a better outcome, and by utilizing therapeutic modalities and behavioral-cognitive tests, assessing psychomotor status, and implementing robotic-based therapies, significant results are obtained [133]. Therefore, all the available therapeutical possibilities have to be used according to the patient’s status for a decrease in morbidity and mortality, as well as the patient’s ability improvement and reintegration into society. In this context, the capacity of music to reconfigure our brains, sharpening various abilities, provides an outstanding avenue as a therapeutic tool in healthcare situations.

## 5. Conclusions

The compendium of research synthesized in this review, titled “Cognitive Crescendo: How Music Shapes the Brain’s Structure and Function”, serves as a seminal contribution to the burgeoning interdisciplinary field at the intersection of musicology, cognitive neuroscience, and clinical psychology. By dissecting a range of subtopics—from rudimentary perceptual features such as pitch, rhythm, and tonality to complex interactions involving emotion, memory, and motor systems—the review offers a comprehensive, integrative framework for understanding how music orchestrates a vast array of neurocognitive processes.

One of the salient contributions of this review is its focus on the bidirectional interactions between music and the limbic system, which has elucidated the underlying neurobiological mechanisms by which music modulates emotional states. The evidence for enhanced connectivity between auditory and emotional regions of the brain brings a new layer of complexity to our understanding of affective regulation and provides fertile ground for future investigation into targeted music-based therapeutic interventions. Regarding motor systems and coordination, the review casts a spotlight on the neural entrainment mechanisms that facilitate synchrony between external rhythmic stimuli and internal neural oscillators. These findings are particularly germane for envisaging music-based rehabilitation paradigms, and the integration of rhythmic elements could revolutionize existing therapeutic approaches.

Furthermore, the review explicates the linguistic dividends of musical training, providing compelling empirical support for shared neural resources between musical and language processing. The implications here are not merely academic but could inform educational curricula that seek to leverage musical training for enhanced linguistic and cognitive skills in children and adults alike. As a corollary to the wide-ranging topics covered, this review also outlines a number of prospective avenues for research. For instance, the operational modifications and neurochemical interactions triggered by chronic exposure to music demand longitudinal studies to ascertain the sustainability of these neural changes. There is also a discernible gap in the literature concerning how these cognitive enhancements translate to real-world skills and well-being, an area ripe for further empirical inquiry.

Another promising avenue for exploration pertains to the therapeutic applications of music. While the existing literature, as summarized in this review, posits a strong case for music as a potent therapeutic tool, the exact protocols, durations, and modalities through which optimal therapeutic outcomes can be achieved remain to be standardized.

In summation, this review serves as both an analytical repository and a conceptual springboard, illuminating the multifaceted ways in which music interacts with the human cognitive apparatus. Its contributions are manifold, offering academic, clinical, and pedagogical insights that advance our understanding of the potent neurocognitive effects of musical engagement. By highlighting nascent areas warranting further exploration, this review not only synthesizes current knowledge but also catalyzes future interdisciplinary research aimed at decoding the myriad ways music intricately shapes our brains and our lives.

## Figures and Tables

**Figure 1 brainsci-13-01390-f001:**
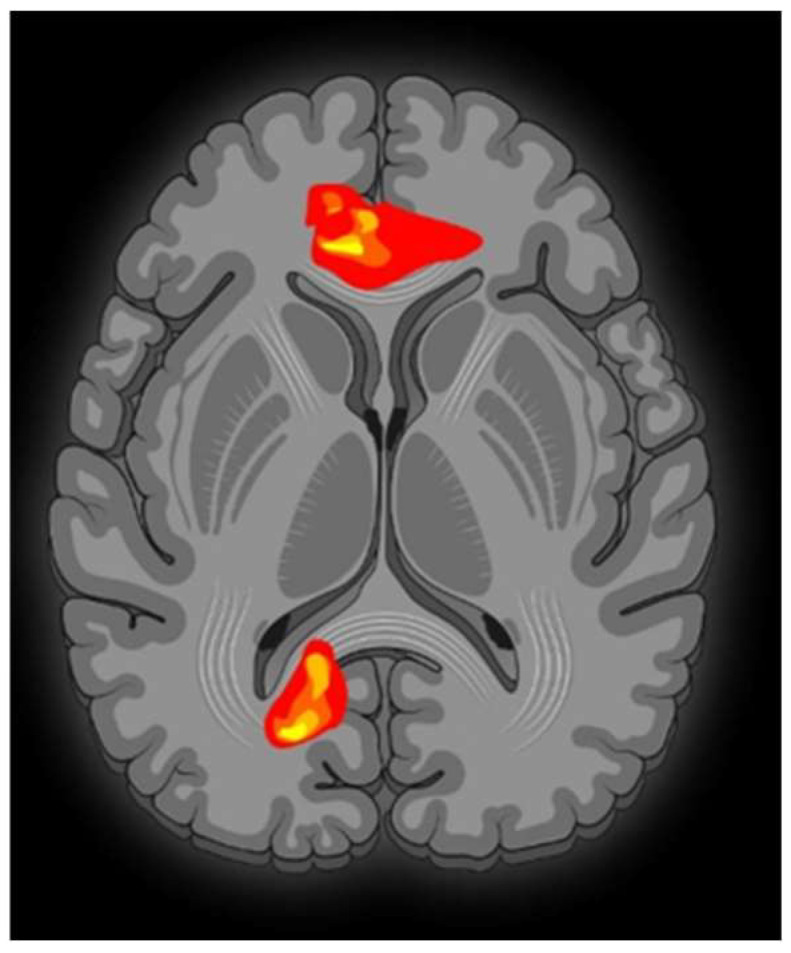
Activation pattern during music listening task. The transversal MRI sequence shows the overall cerebral activation pattern. The lower part of the image will be further explained in Figure 2, while the upper part will be specifically described in Figure 3.

**Figure 2 brainsci-13-01390-f002:**
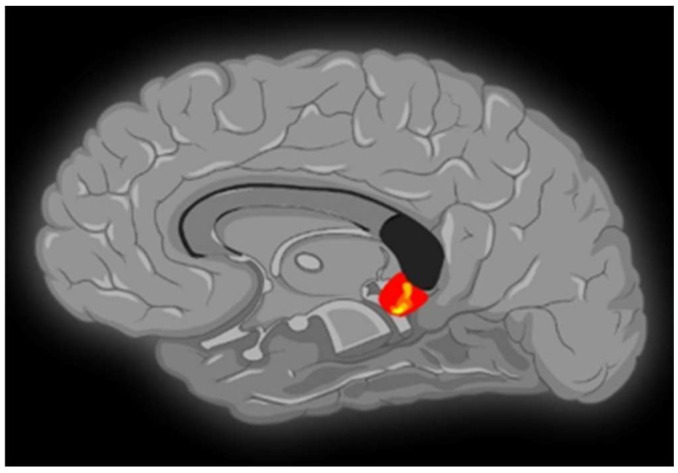
A sagittal MRI sequence is shown, which depicts significant neural activity in the right parahippocampal gyrus.

**Figure 3 brainsci-13-01390-f003:**
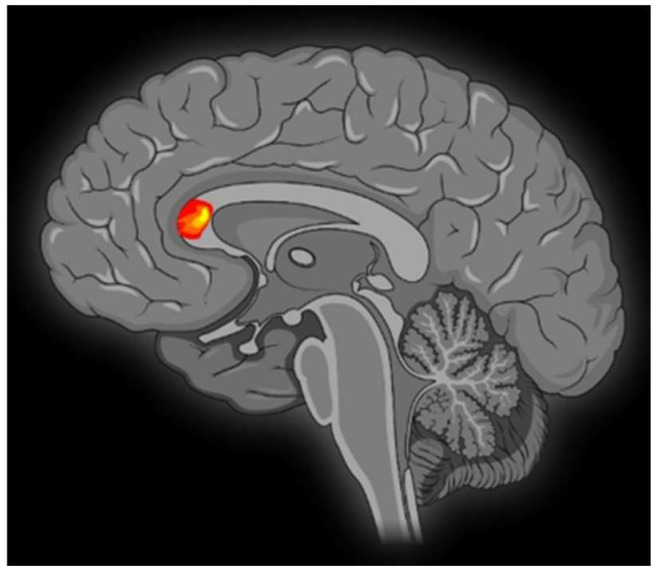
A sagittal MRI sequence is shown, which shows significant neural activity in the right anterior cingulate cortex (BA 24), left anterior cingulate cortex (BA 24), and left medial frontal gyrus (BA 10).

**Table 1 brainsci-13-01390-t001:** Brain areas activated during music listening. Auditory cortices from temporal lobe and limbic system areas are the most frequently implicated brain regions in music processing, as well as other eloquent areas depicted in the table.

	Region (Brodmann Area)
*Temporal cortex*	
Right	Primary auditory cortex (41)
	Secondary auditory cortex (22 and 42)
	Superior temporal sulcus (21 or 22)
	Temporal pole (22 or 38)
	Middle temporal gyrus (21)
Left	Primary auditory cortex (41)
	Superior temporal sulcus (21 or 22)
*Limbic areas*	
Right	Anterior insula
	Hippocampus
Left	Retrosplenial cortex (29 or 30)
	Anterior cingulate cortex (32)
	Anterior insula
	Subcallosal cingulate gyrus (11 or 25)
*Others brain areas*	
	Lingual gyrus (18 and 19)
	Inferior parietal lobule (39)

**Table 2 brainsci-13-01390-t002:** Optimal profile for a patient with high responsiveness to melodic intonation therapy.

Significantly restricted speech ability, or nonfluent speaking
Left-hemisphere stroke, usually unilateral
Possibility to reproduce words while singing well-recognized songs
Moderate integrity of auditory function
Continuously failed attempts to speak
Motivated patients with a great psychological stability
Difficult capacity of repetition

## Data Availability

All data are available online in libraries such as PubMed.

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
