# Peer review of "Cognitive Crescendo: How Music Shapes the Brain’s Structure and Function"

_brainsci, 2023, doi:10.3390/brainsci13101390_

Round 1

Reviewer 1 Report

Title: “Cognitive Crescendo: How Music Shapes the Brain's Structure and Function”

This work is a review about the interaction between music and neuroplasticity. The authors claim that music can architecturally and functionally reconfigure the brain, sharpening various abilities. In addition it has promising therapeutic potentials in healthcare scenarios.

General comment: Although the aim of this review is interesting , this version of the work is clearly not suitable for publication in the journal. Indeed, the format of the main text has some macroscopic errors which has to be corrected to increase the quality and the impact of this contribution. Some examples of very major points to rework:

lines: "Introduction
The inherent complexity of music renders it a multifaceted subject that eludes simple defnitions. While many describe it as an ordered arrangement of sounds, musical elements such as melody, harmony, and bass line require intricate understanding and considerable effort to master. In this research, our focus is on the neurological benefts of music and how the brain responds during varied activities set within a musical context [1].
Music is a universal phenomenon that utilizes a myriad of brain resources. Engaging with music is among the most cognitively demanding tasks a human can undertake, and
its prevalence across all cultures underscores its fundamental human nature [2]. The proclivity to create and appreciate music is ubiquitous among humans, permeating daily life
across diverse societies [3]. This inherent connection to musical expression is deeply intertwined with human identity and experience. Molnar-Szakacs further emphasizes music's unique capacity to evoke memories, stimulate emotions, and enrich social interactions [3]. Historical examples underscore the therapeutic potential of music. For instance,
Johann Sebastian Bach's Goldberg Variations (BWV 988) was purportedly composed to
alleviate a count's insomnia, underscoring music's therapeutic potential [4-6]. The profound emotional impact of music, whether it be the melancholy evoked by a nocturne
from F. Chopin or the elation induced by W. A. Mozart, has inspired ongoing research
into its relationship with emotions and psychological disorders [7]. Fundamental to understanding music are the concepts of pitch perception, rhythm perception, and tonality
perception.
"  etc.....

1) These lines are at the beginning of the text.... as a consequence, this work do not have an abstract. Please add it.

2) The amount of citations (130) should be organized in a logic way. A review paper should describe the current state of the art by viewing the topic following a particular point of view. This is not clear in the current version of this work.

Lines: "2.1. Pitch perception:

2.2. Rhythm perception:

Review of the Literature
In this comprehensive review segment, we delve into existing studies, results, and theoretical postulations regarding the neurological implications of music and its therapeutic applications. The aim is to furnish a meticulous analysis of the current state of knowledge within this feld, accentuating pivotal research endeavors, methodologies, and
discoveries. Subdivisions within this section are delineated based on thematic content,
research domains, or specifc dimensions of the topic.
I. Emotion and Reward Mechanisms in Musical Perception

1. The Interplay of Music with the Limbic System
Central to our emotional resonance with music is the limbic system, an intricate assembly of neural circuits and pathways. Key components of this system, such as the amygdala—responsible for emotional processing—and the hippocampus—integral to memory
consolidation—become activated during musical exposure (
Table 1). Such neural activities account for the evocative power of music to invoke vivid emotional and mnemonic
experiences.

etc etc

3) The numbering of paragraph should follow a logic flow. Normally an increasing oder of the numbering is used. The same number of paragraph should be avoided

Figures 1 ,2, 3

4) The quality of the plots and of the captions should be improved

5) The quality of the language should be improved

The language should be improved

Author Response

Dear Reviewer,

Thank you for your positive feedback and kind suggestions,

We’ve added a comprehensive abstract which explains the subject context and aim of our study

The point of view regarding music-based therapies potential in rehabilitation is more discussed in the current form of the manuscript

We’ve improved the numbering of paragraph:

- “Introduction” explains three key concepts in music studies, those are numbered as “1. Pitch perception”, “2. Rhythm perception” and “3. Tonality perception”.

- “Review of the Literature”, the main body of the manuscript, contains chapters numbered I, II, III and IV, each of them discusses specific subjects in subchapters which are numbered as 1, 2, 3 etc.

- “Conclusions”, the last part of our study, each conclusion is numbered as 1, 2, 3 etc.

The captions were improved accordingly to the plots.

Additionally, Figures 1 and 2 were divided into: Figure 1 (transversal MRI sequence which shows the overall brain activation pattern), respectively Figures 2 and 3 (sagittal MRI sequences which specifically depicts the exact brain areas activated in Figure 1, the lower part is represented in Figure 2, while the upper part is represented in Figure 3)

Quality of language is improved

Thank you for your significant contribution!

Reviewer 2 Report

The presented article provides a number of expert insights and it is evident that the article has been prepared by true experts on the subject. I consider its specific focus on one particular issue to be very professional. I miss the following: 

1. The lack of an abstract: The absence of an abstract is a serious shortcoming of the article. The abstract serves to give readers a quick and concise overview of the article's content and allows them to decide whether they want to read the article in more detail. The lack of an abstract can discourage potential readers.

2. Lack of emphasis on rehabilitation approaches: if the article focuses on the relationship between music and the brain, it is important to include a discussion of how these findings can be applied to the field of rehabilitation. Thus, the lack of emphasis on specific rehabilitation approaches is a major shortcoming, especially if the article was presented as relevant to this issue.

3. I recommend concluding by mentioning the importance of comprehensive rehabilitation approaches: for example: https://www.webofscience.com/wos/woscc/full-record/WOS:000801505500001 

https://www.webofscience.com/wos/woscc/full-record/WOS:000528487500007

The authors have captured the essence of the issue under study and the article thus definitely has a quality impact on the scientific community. I also appreciate the quality focus on the motor system. The comprehensiveness of the article points to a quality resource that I definitely recommend publishing. I only ask the authors for minor changes. 

Author Response

Dear Reviewer,

Thank you for your positive feedback and kind suggestions,

We’ve added a comprehensive abstract which explains the subject context and aim of our study

More observations regarding usage of music in neurodegenerative disorders rehabilitation were added

The final conclusion remark includes the two outstanding studies you’ve mentioned, and is focused on the current status of rehabilitation management

Thank you for your significant contribution!

Round 2

Reviewer 1 Report

Title: “Cognitive Crescendo: How Music Shapes the Brain's Structure and Function”

This work is a review about the interaction between music and neuroplasticity. The authors claim that music can architecturally and functionally reconfigure the brain, sharpening various abilities. In addition it has promising therapeutic potentials in healthcare scenarios.

General comment: The work has been partially revised by the authors. However, some points should be still improved to enhance the overall quality of the work:

1) The particular point of view of the authors about the interaction between music and neuroplasticity is not totally clear. In other words, it is not too clear what is the main contribution given by this work to the field. This important issue should be explained in the first part of the work and the general structure of the text should be shaped accordingly. For instance,  the words :”Our review focused on the neurological and psychological implications of music, as well as presenting the significant clinical relevance of therapies using music.” are not too clear, since the authors described “neurological and psychological implications” of listening music, since music is “per se” a complex acoustic phenomenon, while listening music by patients has neurological and psychological implications. Please clarify and improve.

2) The numbering of paragraphs and subparagraphs should be improved. Please use a single style. For instance : Paragraph 1, Subsection 1.1, Subsection 1.2, etc… Paragraph 2, Subsection 2.1, Subsection 2.2, etc..

2.1) Do not mix different kind of numbering e.g., I,II,II, 1,2,3 etc. Again, II, a), b) etc. Please avoid lists a),b),c) if not strictly needed.

2.2) Also lines as “Accurate Auditory Processing: This entails discerning minute time deviations and provides the bedrock for rhythm perception, enabling the detection of intricate temporal 36patterns. Brain OscillationSynchronization: This mechanism concerns the brain's ability to anticipate ensuing events and conform to hierarchical rhythm structures. It ensures alignment of rhythmic components, contributing to the holistic rhythm experience [47]. Sensorimotor Coupling: This establishes a link between perception and execution, implicating the motor system”, etc.. are not clear. Please avoid lists of bold keywords (Accurate Auditory Processing, Brain Oscillation Synchronization:,Sensorimotor Coupling) ,which are not clearly inserted within the main text.

3) “Conclusion” paragraph: it should be better shaped to clarify to the readers the particular point of view of the authors. Again the main question: what is the main contribution of this review to the state of the art ? Please explain in a better way.

4) The overall structure of the work is not totally clear: although this is a review work, where is the “Methods” section ? What criteria were used by authors to find relevant literature ? What is the main aim (again)… and how this aim influenced the criteria of choice ? Please rework and clarify.

The language can be improved: the logic flow of the arguments can be also improved

Author Response

Thank you for taking the time to thoroughly review our manuscript entitled “Cognitive Crescendo: How Music Shapes the Brain's Structure and Function.”

We appreciate the constructive feedback and are keen to address your concerns to improve the overall quality of the work.

1) Clarity on the Contribution to the Field: We acknowledge your concern regarding the specificity of our point of view and its contribution to the field. In light of this, we have revised the introduction to more explicitly state our unique contribution. We aimed to emphasize that our review synthesizes interdisciplinary perspectives to underscore the neurological, psychological, and therapeutic implications of music. Your observation about the distinction between music as an acoustic phenomenon and its neurological and psychological impact is particularly helpful; we have refined the text to clarify these aspects.

2) Uniform Numbering and Structure: Thank you for pointing out the inconsistency in the numbering and structure of paragraphs and subsections. We have implemented a uniform numbering system across the text to enhance readability and organizational flow (e.g., Paragraph 1, Subsection 1.1, Subsection 1.2, etc.).

    2.1) Consistent Numbering Style: We appreciate this detailed observation and have employed a single style of numbering through the text.

    2.2) Use of Bold Keywords: We understand your concern about the clarity of lines containing bold keywords like “Accurate Auditory Processing,” “Brain Oscillation Synchronization,” and “Sensorimotor Coupling.” We have revised the use of this kind of words.

3) Improving the Conclusion: Your point about refining the conclusion to make our unique contribution more transparent is well-taken. We have revised this section to summarize our unique viewpoint, emphasizing the specific contributions and implications of our work for the current state of the art in this interdisciplinary field.

4) We appreciate the opportunity to clarify the structural conventions associated with our manuscript, which is formatted as a narrative review. In the academic tradition, narrative reviews are generally not accompanied by a "Methods" section that outlines explicit procedures for the systematic selection, inclusion, or exclusion of articles. Rather, the narrative review is characterized by its qualitative synthesis of relevant literature, aimed at providing a comprehensive overview and critical discussion of key themes and findings within a given field. In this context, our objective has been to aggregate and interpret a broad array of research contributions in a manner that offers cohesive insights into the neurological, psychological, and therapeutic implications of music. We trust this elucidation serves to clarify the methodological approach underlying our review

We hope these revisions will address your concerns effectively. Thank you once again for your valuable input, which provides us with an excellent opportunity to refine and improve our manuscript.

With high regard,

The collective of authors